# Generalize to Fully Unseen Graphs: Learn Transferable Hyper-Relation Structures for Inductive Link Prediction

## ABSTRACT

Inductive link prediction aims to infer missing triples on unseen graphs, which contain unseen entities and relations during training. The performances of existing inductive inference methods were hindered by the limited generalization capability in fully unseen graphs, which is rooted in the neglect of the intrinsic graph structure. In this paper, we aim to enhance the model's generalization ability to unseen graphs and thus propose a novel **Hy**per-**Re**lation aware multi-views model (**HyRel**) for learning the global transferable structure of graphs. Distinct from existing studies, we introduce a novel perspective focused on learning the inherent hyper-relation structure consisting of the relation positions and affinity. The hyper-relation structure is independent of specific entities, relations, or features, thus allowing for transferring the learned knowledge to any unseen graphs. We adopt a multi-view approach to model the hyper-relation structure. HyRel incorporates neighborhood learning on each view, capturing nuanced semantics of relative relation position. Meanwhile, dual views contrastive constraints are designed to enforce the robustness of transferable structural knowledge. To the best of our knowledge, our work makes one of the first attempts to generalize the learning of hyper-relation structures, offering high flexibility and ease of use without reliance on any external resources. HyRel demonstrates SOTA performance compared to existing methods under extensive inductive settings, particularly on fully unseen graphs, and validates the efficacy of learning hyper-relation structures for improving generalization. The code is available online at https://github.com/hncps6/HyRel.

## CCS CONCEPTS

• **Computing methodologies** → **Knowledge representation and reasoning**.

## KEYWORDS

Inductive Link Prediction; Representation Learning; Knowledge Embedding

## 1 INTRODUCTION

Link prediction task [15] aims to predict missing triples in the knowledge graph, where each triple consists of a head entity $h$, a relation $r$, and a tail entity $t$, denoted as $(h, r, t)$. For example, predicting the missing head entity $h$ in the triple $(?, r, t)$ or the missing

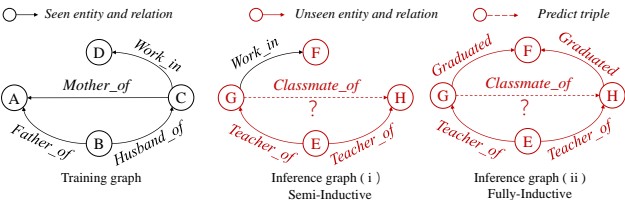

(a): Examples of inductive inference.

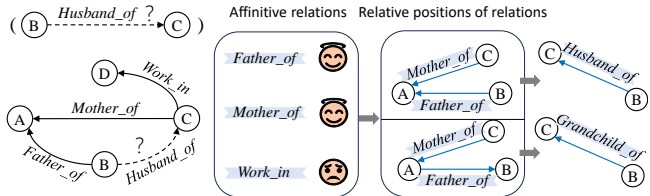

(b): Illustration of inference process.

**Figure 1: Simple examples of inductive inference(a) and the inference process(b).**

tail entity $t$ in the triple $(h, r, ?)$. Methods using low-dimensional embedding vectors [6, 31] have proven effective for link prediction. Entities and relations are transformed into embedding vectors, enabling inference of missing triples. However, these methods [2, 8, 19, 27, 36] follow the transductive setting, i.e. require entities and relations to be present in the training set. When entities or relations unseen during training occur, they no longer work.

Inductive link prediction aims to infer missing triples on unseen graphs, which contain unseen entities and relations during training, even fully unseen. The unseen entities and relations mean they are not present in the training set. As shown in Fig.1(a), inference graph (i) and (ii) contain unseen entity $G$ haven't seen in training graph. All entities and relations in inference graphs (ii) are not seen in the training graph. An example of inductive link prediction is also shown in inference graph (ii), e.g. infer missing tail entity $H$ in the triple $(G, Classmate\_of, ?)$. In inductive link prediction, there are two scenarios [10, 14] of inductive learning: (i) fully unseen entity set with partially unseen relations set during inference (semi-inductive link prediction) and (ii) fully unseen entity set with fully unseen relations set (full-inductive link prediction). Inference graph (i) and inference graph (ii) in Fig.1(a) show these scenarios.

A variety of inductive link prediction methods have been developed. Some works attempt to obtain embeddings for unseen entities from pre-trained language models or textual descriptions, or propagate information from seen entities to unseen entities through graph neural networks [5, 11, 17, 29]. However, additional textual descriptions or explicit associations between unseen entities and seen entities may not always be available in practical application scenarios. In addition, the approaches of learning transferable features through meta-learning or other subgraph partitioning methods [4, 10], while not relying on external resources, lack access to

the global structural information of the given knowledge graph. Recently, INGRAM[14] proposed to employ relation-level aggregation methods to address full inductive link prediction. The performances of such methods were hindered by the limited generalization capability.

To address the above issue, we tend to learn transferable knowledge rather than rely on specific entities or relations, which is similar to the way humans think. Take the scenario in Fig.1(b) as an example, when inferring the triple $(B, Husband\_of, C)$, relevant information could be extracted from its associated relational structure, such as $Mother\_of$ and $Father\_of$, while other relations such as $Work\_in$ may be less relevant to the inference. This means that the relation $Husband\_of$ is more affinitive with $Mother\_of$ and $Father\_of$ than with $Work\_in$ in this inference. After identifying the affinitive relations, it's crucial to assess their relative positions. Specifically, if the relations $Mother\_of$ and $Father\_of$ both directed towards entity $A$, it could be inferred that $(B, Husband\_of, C)$ holds true. Alternatively, if $Mother\_of$ and $Father\_of$ are not both directed towards entity $A$, and the direction of $Father\_of$ is not known, $B$ could be either the $husband$ or the $grandchild$ of $C$.

The core of the inference process outlined above is the recognition of affinity relations and the relative positions of these relations. In the inference graph (ii) of Fig.1(a), the validity of $(G, Classmate\_of, H)$ could be determined according to the affinitive relations $Teacher\_of$ and $Graduated$ with their relative directions. We refined affinitive relations and relative positions of relations, which are independent of any specific entity, relation, or graph, as **Hyper-Relation Structure**. It could be considered as transferable knowledge because it naturally exists in any graph. We argue that if the model learns this hyper-relation structure during training, it will enable the model to make inferences on unseen graphs, i.e., achieve semi-inductive link prediction and full inductive link prediction.

To this end, we propose a novel **Hy**per-**Rel**ation aware multi-views approach for inductive link prediction (HyRel). It operates independently of pre-trained models or external resources, which provides high flexibility and ease of use. Specifically, we adopt a multi-view approach to model the hyper-relation structure, where different relative positions of relations form different views. In each view, relations are treated as nodes, and the affinity (the modeling of affinity between relations is formally defined in Section 4.1) between relations serves as edges. HyRel learns fine-grained semantics of relation contexts at different positions within each view and generalizes this learning to unseen graphs. Furthermore, we propose dual views contrastive constraints to reduce the impact of local information confusion, enhancing the robustness of learned intrinsic and transferable structural knowledge. To the best of our knowledge, our work makes one of the first attempts to generalize the learning of hyper-relation structures. Specifically, our contributions are summarized as follows:

- Distinct from existing studies, we introduce a novel perspective focused on learning the hyper-relation structure inherent in graphs, allowing for transferring the learned knowledge to any unseen graphs.
- We are the first to incorporate the affinity of different relative positions of relations into inductive link prediction. By leveraging

the semantic differences between different relative positions, we aggregate more refined semantic information.
- We propose dual views contrastive constraints to alleviate semantic confusion of relations and enforce the robustness of transferable structural knowledge.

The experimental results in various inductive settings demonstrate that HyRel outperforms existing models in inductive link prediction, particularly excelling in challenging fully inductive tasks where entities and relations are entirely unseen.

## 2 RELATED WORK

**Inductive link prediction with associated seen entities.** Early exploration into inductive link prediction could be traced back to [13] and [29]. They require these unseen entities to be associated with seen entities in the training set. Therefore, such methods cannot be generalized to settings where entities are entirely unseen. **Subgraph-based methods.** These methods [20, 21, 25, 33] extract local subgraphs and employ GNN modules [3, 23] to enable relation prediction. While these methods could handle scenarios involving unseen entities, they cannot generalize to settings involving the emergence of unseen relations. [10] also falls into this category but could operate on inference graphs with unseen relations. The task handled by [10] is similar to ours; however, its method of extracting local subgraphs for each candidate entity to compute scores does not calculate embedding vectors, rendering it inapplicable for downstream tasks.

**Rule-guided and path formulation.** These approaches aim to discover common logical rules or relation paths in graphs. As logical rules are not specific to particular entities, such methods could be generalized to handle inductive tasks involving unseen entities. Both [35] and [22] integrate parameter and structure learning of first-order logical rules into end-to-end differentiable models. [38] introduces a general framework for path representation learning, proposing new operators for aggregating node pairs and path embeddings. [37] constructs more complex relational digraphs than paths to capture local evidence. **Language Models.** Methods like [1, 7, 11] encode the textual descriptions of each entity using pre-trained language models. These approaches utilize the corresponding text descriptions to obtain embedding for unseen entities. In our work, embeddings for entities and relations are solely derived from the graph structure, without relying on any external sources, aiming to be resource-efficient and generalizable.

**A method similar to ours.** INGRAM[14] is similar to ours, as both approaches could handle the scenario of complete inductive link prediction, where entities and relations could be entirely unseen. INGRAM was the first to introduce the concept of affinity. However, it measures the affinity between two relations by the sum of the counts of shared head entities and shared tail entities. In addition, it overlooks the significant differences in affinity between different relative positions of relations. In other words, both the positional information and affinity between each relation should be considered simultaneously, as the affinity between relations varies depending on different relative positions. We comprehensively consider the relative positional information between relations and the affinity, aggregating fine-grained relational semantic information

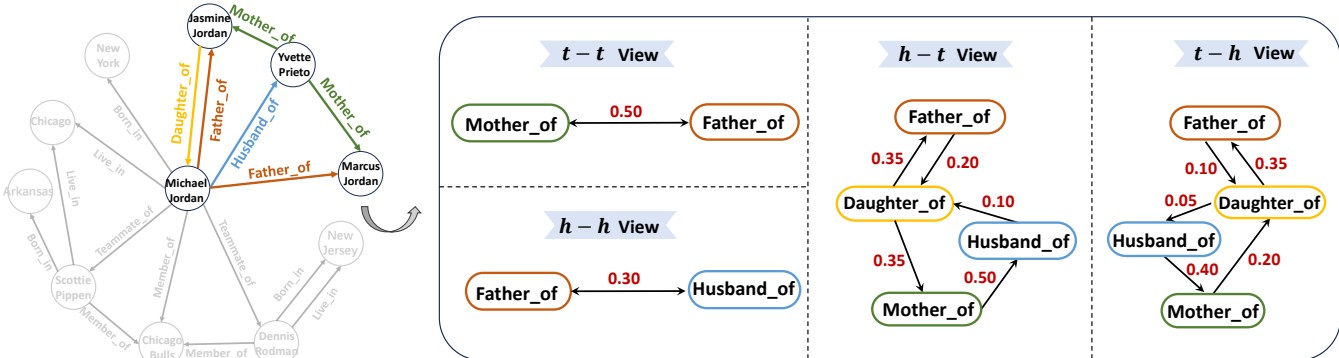

**Figure 2: The figure shows the importance of the hyper-relation structure.** *Mother_of* and *father_of* have a high affinity in the *t-t* view, but these two relations do not appear in the *h-h* view because entities cannot be both the father and the mother.

by enhancing the distinctiveness of each relative positional information(details will be provided in Section 4.2).

## 3 PRELIMINARIES

Next, we will explain the setup of the current task in this paper. The training graph is defined as $\widetilde{\mathcal{G}_{\text{tra}}} = (\mathcal{E}_{\text{tra}}, \mathcal{R}_{\text{tra}}, \mathcal{F}_{\text{tra}})$, where $\mathcal{E}_{\text{tra}}$ is the set of entities, $\mathcal{R}_{\text{tra}}$ is the set of relations, and $\mathcal{F}_{\text{tra}}$ is the set of triples in the training graph $\widetilde{\mathcal{G}_{\text{tra}}}$. We divide $\mathcal{F}_{\text{tra}}$ into $\mathcal{F}_{\text{sup}}$ and $\mathcal{F}_{\text{opt}}$, such that $\mathcal{F}_{\text{tra}} = \mathcal{F}_{\text{sup}} \cup \mathcal{F}_{\text{opt}}$. $\mathcal{F}_{\text{sup}}$ is a set of seen factual triples aimed at obtaining representations for all entities and relations in the training graph, while $\mathcal{F}_{\text{opt}}$ is a set of factual triples used to calculate triple scores for the purpose of optimizing the model. $\widetilde{\mathcal{G}_{\text{inf}}}$ is the inference graph, defined as $\widetilde{\mathcal{G}_{\text{inf}}} = (\mathcal{E}_{\text{inf}}, \mathcal{R}_{\text{inf}}, \mathcal{T}_{\text{inf}})$, where $\mathcal{E}_{\text{inf}}$ represents the set of entities, $\mathcal{R}_{\text{inf}}$ represents the set of relations, and $\mathcal{T}_{\text{inf}}$ represents the set of triples in the inference graph $\widetilde{\mathcal{G}_{\text{inf}}}$. We divide $\mathcal{T}_{\text{inf}}$ into three parts, $\mathcal{T}_{\text{sup}}$, $\mathcal{T}_{\text{val}}$, and $\mathcal{T}_{\text{tes}}$, such that $\mathcal{T}_{\text{inf}} = \mathcal{T}_{\text{sup}} \cup \mathcal{T}_{\text{val}} \cup \mathcal{T}_{\text{tes}}$. During testing, $\mathcal{T}_{\text{sup}}$ is used to obtain embeddings for entities and relations in the inference graph. $\mathcal{T}_{\text{val}}$ represents the validation set of the inference graph, and $\mathcal{T}_{\text{tes}}$ represents the test set. After computing the embeddings for entities and relations using $\mathcal{T}_{\text{sup}}$, the corresponding embeddings are obtained for $\mathcal{T}_{\text{val}}$ and $\mathcal{T}_{\text{tes}}$ to perform validation and testing, respectively.

We set the ratio of $\mathcal{F}_{\text{sup}}$ to $\mathcal{F}_{\text{opt}}$ in the training graph as 3:1, and the ratio of $\mathcal{T}_{\text{sup}}$, $\mathcal{T}_{\text{val}}$, and $\mathcal{T}_{\text{tes}}$ in the inference graph as 3:1:1, by adapting the model to different graph structures through the transformation of triplets in $\mathcal{F}_{\text{sup}}$ and $\mathcal{F}_{\text{opt}}$ within each epoch during training, following the settings of previous work [1, 9, 14]. Entities in the inference graph are unseen to the training graph, meaning that all entities in the inference graph are new entities, i.e., $\mathcal{E}_{\text{tra}} \cap \mathcal{E}_{\text{inf}} = \emptyset$. Furthermore, Our method can handle both cases where partial relations are unseen, i.e., $\mathcal{R}_{\text{tra}} \neq \mathcal{R}_{\text{inf}}$, and where relations are completely unseen, i.e., $\mathcal{R}_{\text{tra}} \cap \mathcal{R}_{\text{inf}} = \emptyset$. Inverse triples [24] induced by inverse relations are also encompassed within our model.

## 4 METHOD

Fig.3 presents the architecture of HyRel. Specifically, (1) hyper-relation structure is extracted from the global structure of the given graph. Then, we adopt a multi-view approach to model the hyper-relation structure, where different relative positions of relations form different views. (2) Hyper-relation structure attention (HyRel-Gat) mechanism is proposed for generating high-quality relation embedding. (3) We impose dual views contrastive constraints across views of hyper-relation structure to alleviate semantic confusion of relations.

## 4.1 Defining Hyper-Relation Structure

We define hyper-relation structure by extracting structural information from the original graph. Firstly, we divided four views based on the distinct relative positions between relations. In other words, each view represents different relative positions between relations. Secondly, we computed the frequency of shared entities between relations at different relative positions and defined it as affinity. This definition is effective because affinitive relations tend to cluster around entities, as shown in Fig.2 with kinship relations consistently clustering together. Finally, each view treats relations as nodes and the affinity as edges. Additionally, the affinity between the same relation pairs varies across different views.

To obtain the four views, we first construct two matrices $E_h \in \mathbb{R}^{n \times m}$ and $E_t \in \mathbb{R}^{n \times m}$, where $n$ represents the number of entities and $m$ represents the number of relations. These matrices indicate the frequency of each entity appearing as a head entity and tail entity across all relations. For example, $E_h[e_1, r_2]$ represents the frequency of entity $e_1$ appearing as the head entity for relation $r_2$ in the original graph. Then, $D_h \in \mathbb{R}^{n \times n}$ is the diagonal matrix representing the degree of entities as head entities, i.e., $D_h[i, j] = \sum_j E_h[i, j]$. Similarly, $D_t \in \mathbb{R}^{n \times n}$ is the diagonal matrix representing the degree of entities as tail entities. Finally, we define the adjacency matrices of the four views as:

$$A_{p_1\text{-}p_2} = \frac{E_{p_1}^T E_{p_2}}{D_{p_1} D_{p_2}} \quad (1)$$

where $p_1\text{-}p_2 \in \{h\text{-}h, t\text{-}t, h\text{-}t, t\text{-}h\}$. For example, $a_{ij|h-t} \in A_{h-t}$ represents the affinity between relation $r_i$ and relation $r_j$ at the $h$-$t$ position, $h$-$t$ refers to the case where the head entity connected by relation $r_1$ is the tail entity of relation $r_2$. Through the above definition, four views $\{\mathcal{G}_{h-h}, \mathcal{G}_{t-t}, \mathcal{G}_{h-t}, \mathcal{G}_{t-h}\}$ could be obtained.

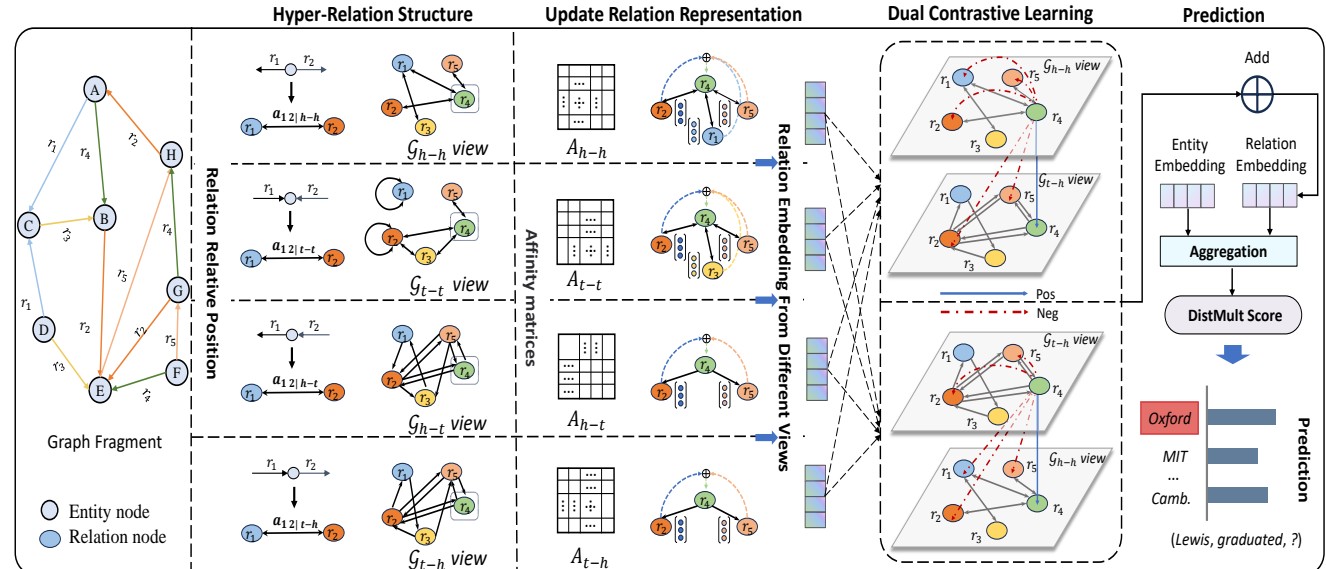

**Figure 3: The overall architecture of the HyRel model. HyRel follows the end-to-end paradigm. Given a graph, four views of the hyper-relation structure are extracted from the original graph. Utilizing these four views, relation node embeddings are derived for each view. Following that, the dual views contrastive constraints module is employed to alleviate semantic confusion of relations. After obtaining relation and entity embeddings in this manner, scores are computed using the DistMult method.**

The connections within each view are established based on the affinity within that specific view. Fig.2 shows an example.

## 4.2 HyRelGat: Update Relation Representation

We use Glorot initialization [12] for initializing relation embeddings, denoted as $\mathbf{v}_i \in \mathbb{R}^{dr}$, where $i$ represents the $i$-th relation, and $dr$ represents the dimension of the relation vectors. Then, the embeddings of relations are updated using the four views. Each relation in the four views obtains four different sets of neighboring relations and their corresponding affinity weights. We define the process of updating relations as follows:

$$\mathbf{x}_i^{(l+1)} = \sigma\left(\left(\sum_{p \in \mathcal{P}} \sum_{r_j \in \mathcal{N}_i^p} \alpha_{ij|p}^{(l)} \mathbf{W}_p^{(l)} \mathbf{x}_j^{(l)}\right) + \mathbf{K}^{(l)} \mathbf{x}_i^{(l)}\right) \quad (2)$$

where $(l)$ represents the $l$-th layer with $l \in \{0, 1, \cdots, L-1\}$. $\mathbf{x}_i^{(0)}$ = $\mathbf{W}_{(rel)}\mathbf{v}_i$, $\mathbf{W}_{(rel)} \in \mathbb{R}^{dr \times dr}$ is a learnable matrix, and $dr'$ is the hidden dimension. $\mathcal{P} = \{\mathcal{G}_{h-h}, \mathcal{G}_{t-t}, \mathcal{G}_{h-t}, \mathcal{G}_{t-h}\}$ represents the set of four different views. $\alpha_{ij|p}^{(l)}$ and $\mathbf{W}_p^{(l)} \in \mathbb{R}^{dr' \times dr'}$ represent four different sets of relative attention parameters and weight matrices, respectively. $\mathcal{N}_i^p$ indicates the set of neighbors for relation $r_i$ at different relative positions. In order to fully leverage the hidden representations at each layer, residual connections are also employed, the output of each layer is transformed by a weight matrix and then passed on to the next layer. $\mathbf{K}^{(l)} \in \mathbb{R}^{dr' \times dr'}$ represents the weight matrix and $\sigma(\cdot)$ represents the activation function.

Through concatenating relation feature vectors, we capture semantic information of relations at different relative positions by

avoiding weight matrix sharing. The specific process is as follows:

$$c_{ij|p}^{(l)} = \mathbf{H}_p^{(l)} \left[\mathbf{x}_i^{(l)} \| \mathbf{x}_j^{(l)}\right] \quad (3)$$

where $\|$ represents concatenation, $\mathbf{H}_p^{(l)} \in \mathbb{R}^{dr' \times 2dr'}$ represents the weight matrix of different views. Next, we define the absolute attention values $b_{ij|p}^{(l)}$ to represent the importance of each triple in different views, and we add learnable affinity parameters into the absolute attention values. Specifically, the calculation process of $b_{ij|p}^{(l)}$ as follows:

$$b_{ij|p}^{(l)} = \left(\varpi^{(l)} \sigma\left(c_{ij}^{(l)}\right) + o_{(i,j)|p}\right) \quad (4)$$

where $o_{(i,j)|p}$ is the learnable parameter, and it is selected based on the ranking of $a_{ij}$ within $\mathbf{A}_{p_1-p_2}$, where a higher ranking indicates a stronger affinity. $\varpi^{(l)} \in \mathbb{R}^{1 \times dr'}$ is a weight vector. Eq.(5) provides a detailed definition of the relative attention coefficients $\alpha_{ij|p}^{(l)}$:

$$\alpha_{ij|p}^{(l)} = \frac{\exp\left(b_{ij|p}^{(l)} + t_p\right)}{\sum_{r_{j'} \in \mathcal{N}_i^p} \exp\left(b_{ij'|p}^{(l)} + t_p\right)} \quad (5)$$

where $t_p$ is the views-adaptive parameter. Due to the differing structures among the four views, the relative attention coefficients should be different accordingly and correspond to the structures of the four views. For instance, if the relative attention coefficient $\alpha_{ij|h-h}^{(l)}$ obtained in $\mathcal{G}_{h-h}$ is equal to $\alpha_{ij|t-t}^{(l)}$ in $\mathcal{G}_{t-t}$, HyRel could differentiate them using $t_p$.

In contrast to current inductive link prediction methods [4, 11, 17, 29], we consider the different importance of relations across different relative positions. Specifically, each of the four views in HyRel has its own independent set of weight parameters: $\mathbf{W}_p$,

$H_p$, and views-adaptive parameter $t_p$. Hence, HyRel can aggregate more fine-grained semantic information by the differences between different views. Then, the final relation embeddings are computed based on the Eq.(2), denoted as $\mathbf{x}_i = M_r \mathbf{x}_i^{(L)} (i = 1, \cdots, m)$, where $M_r \in \mathbb{R}^{dr \times dr'}$ is a learnable mapping matrix.

## 4.3 Dual Views Contrastive Constraints

We design a novel contrastive learning, which imposes contrastive constraints on four views. Specifically, unlike the construction of negative sample pairs in common graph constrastive learning [16, 18, 30], here we only consider the neighbors of nodes as negative pairs to avoid relational semantic confusion. In other words, the anchor's neighbors within the same view and across different views are treated as negative pairs.

For instance, we define $\mathbf{x}_{i|h-h}^{(L)}$ and $\mathbf{x}_{i|t-h}^{(L)}$ as the L2-normalized embeddings of relation $r_i$ learned from the $h$-$h$ view and $t$-$h$ view, then $\mathbf{x}_{i|h-h}^{(L)}$ serving as the anchor. In addition, positive samples are the same nodes across different views, and negative samples are: (1) nodes of neighbors within the $h$-$h$ view, i.e., $\left\{ \mathbf{x}_{j|h-h}^{(L)} | r_j \in \mathcal{N}_i^{h-h} \right\}$, and (2) nodes of neighbors within the $t$-$h$, i.e., $\left\{ \mathbf{x}_{j|t-h}^{(L)} | r_j \in \mathcal{N}_i^{t-h} \right\}$. The contrastive loss for $\mathbf{x}_{i|h-h}^{(L)}$ between the $h$-$h$ and $t$-$h$ views is formulated as follows:

$$\ell \left( \mathbf{x}_{i|h-h}^{(L)}, \mathbf{x}_{i|t-h}^{(L)} \right) = -\log \frac{\mathbf{sim}_{(pos)}}{\mathbf{sim}_{(pos)} + \mathbf{sim}_{(neg)}} \quad (6)$$

where $\mathbf{sim}_{(pos)}$ represents the similarity between positive pairs, and is defined as:

$$\mathbf{sim}_{(pos)} = e^{\theta \left( \mathbf{x}_{i|h-h}^{(L)}, \mathbf{x}_{i|t-h}^{(L)} \right)/\tau} \quad (7)$$

And $\mathbf{sim}_{(neg)}$ represents the similarity between negative pairs:

$$\mathbf{sim}_{(neg)} = \sum_{r_j \in \mathcal{N}_i^{h-h}} e^{\theta \left( \mathbf{x}_{i|h-h}^{(L)}, \mathbf{x}_{j|h-h}^{(L)} \right)/\tau} + \sum_{r_j \in \mathcal{N}_i^{t-h}} e^{\theta \left( \mathbf{x}_{i|h-h}^{(L)}, \mathbf{x}_{j|t-h}^{(L)} \right)/\tau} \quad (8)$$

where $\theta(\cdot)$ is a similarity metric function(default inner product). $\tau$ is a temperature hyperparameter used to control the scale of the similarity.

In the overall dual views contrastive constraints , the $h$-$h$ view serves as the central view and is contrastive with the other three views, i.e., $\ell \left( \mathbf{x}_{r|h-h}^{(L)}, \mathbf{x}_{r|t-h}^{(L)} \right), \ell \left( \mathbf{x}_{r|h-h}^{(L)}, \mathbf{x}_{r|t-t}^{(L)} \right)$ and $\ell \left( \mathbf{x}_{r|h-h}^{(L)}, \mathbf{x}_{r|h-t}^{(L)} \right)$. In Eq. (6), we explained the loss formulation with $r_i$ from the $h$-$h$ view as the anchor. Similarly, when $r_i$ from the other views is used as the anchor, the contrastive loss can be obtained in the same manner. To perform pairwise comparisons, the final contrastive loss is defined as follows:

$$\mathcal{L}_c = \frac{\sum_{p \in \mathcal{P}\prime} \sum_{i=1}^{N} \left[ \ell \left( \mathbf{x}_{i|h-h}^{(L)}, \mathbf{x}_{i|p}^{(L)} \right) + \ell \left( \mathbf{x}_{i|p}^{(L)}, \mathbf{x}_{i|h-h}^{(L)} \right) \right]}{2|\mathcal{P}\prime| N} \quad (9)$$

where $\mathcal{P}\prime = \{ \mathcal{G}_{t-t}, \mathcal{G}_{h-t}, \mathcal{G}_{t-h} \}$. Section 5.6 explains that the selection of the central view has a minimal impact on performance.

## 4.4 Update The Entity Representation Vectors

We use Glorot initialization [12] for initializing entity embeddings, denoted as $\mathbf{e}_i \in \mathbb{R}^{de}, i = 1, \cdots, n$, where $de$ is the dimension of the entity vectors. Then, similar to updating relation embeddings, the entity embeddings are updated by aggregating neighboring entities through a multi-head attention mechanism and residual connections.

We define $\mathbf{z}_i^{(l)} \in \mathbb{R}^{de'}$ as the hidden representation of entity $e_i$, where $(l)$ represents the $l$-th layer with $l \in \{0, 1, \cdots, \widetilde{L} - 1\}$ and $de'$ is the hidden dimension. $\mathbf{z}_i^{(0)} = W_{(ent)} \mathbf{e}_i$, where $W_{(ent)} \in \mathbb{R}^{de' \times de}$ is a learnable matrix. The specific process for updating entity embeddings is as follows:

$$\mathbf{z}_i^{(l+1)} = \sigma \left( \sum_{e_j \in \widetilde{\mathcal{N}}_i} \sum_{r \in \mathcal{R}_{ji}} \widetilde{\alpha}_{ijr}^{(l)} \widetilde{W}^{(l)} \left[ \mathbf{z}_j^{(l)} \| \mathbf{x}_r^{(L)} \right] \right) \quad (10)$$

where $\mathcal{R}_{ji}$ is the set of relations from entity $i$ to $j$, $\widetilde{W}^{(l)} \in \mathbb{R}^{de' \times de'}$ is the weight matrix. $\widetilde{\alpha}_{ijr}^{(l)}$ is defined as the entity-level attention coefficient, which is computed by concatenating the feature vectors of neighboring entities and connected relations for each entity. To compute the attention weight for the self-loop of entities. We utilize the mean vector of the representation vectors of the relations adjacent to entities, which is similar to the strategies adopted in [14]. The mean embedding $\overline{\mathbf{x}}_r^{(L)}$ is concatenated with $\mathbf{z}_i^{(l)}$ and $\mathbf{z}_i^{(l)}$ to calculate the self-loop attention weight of the entity $i$, i.e., $\left[ \mathbf{z}_i^{(l)} \| \mathbf{z}_i^{(l)} \| \overline{\mathbf{x}}_i^{(L)} \right]$.

The final representation of the entity is $\mathbf{z}_e = M_e \mathbf{z}_i^{(\widetilde{L})} (i = 1, \cdots, n)$, where $M_e \in \mathbb{R}^{de \times de'}$ is a learnable mapping matrix.

## 4.5 Model Learning

The model is trained to ensure that positive triples in $\mathcal{F}_{opt}$ obtain higher scores compared to the sampled negative triples. We have employed a variant of the DistMult model [34] as the scoring function. The scoring function is defined as $f \left( e_i', r, e_j' \right) = \mathbf{z}_i^T diag(M_{re} \mathbf{x}_r) \mathbf{z}_j$, where $r = (1, \cdots, m)$, $M_{re} \in \mathbb{R}^{de \times dr}$ is a weight matrix and $diag(M_{re} \mathbf{x}_r)$ represents a diagonal matrix. With this scoring function, we could derive a loss function specific to the link prediction.

$$\mathcal{L}_g = \sum_{(e_i, r, e_j) \in \mathcal{F}_{opt}} \sum_{(e_i', r, e_j') \in \mathcal{F}_{opt}'} \max \left( 0, \gamma - f \left( e_i, r, e_j \right) + f \left( e_i', r, e_j' \right) \right) \quad (11)$$

$\mathcal{F}_{opt}'$ is a set of negative triples, which are generated by altering the head or tail entity of positive triples. The parameter $\gamma$ serves as a threshold to differentiate the margin between positive and negative triples. By combining the contrastive constraints loss, we can obtain a composite loss function for optimizing our model:

$$\mathcal{L} = \mathcal{L}_c + \mathcal{L}_g \quad (12)$$

**Table 1: The performance of HyRel and baselines in inductive link prediction is evaluated on twelve datasets. The number following each dataset represents the proportion of new relations. * represents the results obtained by replicating the model on these datasets. + represents that we replicated the model, and for a fair comparison, we selected the best results reported in the paper.**

| Model | NL-100 | | | NL-75 | | | NL-50 | | | NL-25 | | |
|---|---|---|---|---|---|---|---|---|---|---|---|---|
| | MRR | Hit@10 | Hit@1 | MRR | Hit@10 | Hit@1 | MRR | Hit@10 | Hit@1 | MRR | Hit@10 | Hit@1 |
| GraIL | 0.135 | 0.173 | 0.114 | 0.096 | 0.205 | 0.036 | 0.162 | 0.288 | 0.104 | 0.216 | 0.366 | 0.160 |
| CoMPILE | 0.123 | 0.209 | 0.071 | 0.178 | 0.361 | 0.093 | 0.194 | 0.330 | 0.125 | 0.189 | 0.324 | 0.115 |
| SNRI | 0.042 | 0.064 | 0.029 | 0.088 | 0.177 | 0.040 | 0.130 | 0.187 | 0.095 | 0.190 | 0.270 | 0.140 |
| INDIGO | 0.160 | 0.247 | 0.109 | 0.121 | 0.156 | 0.098 | 0.167 | 0.217 | 0.134 | 0.166 | 0.206 | 0.134 |
| RMPI | 0.220 | 0.376 | 0.136 | 0.138 | 0.275 | 0.061 | 0.185 | 0.307 | 0.109 | 0.213 | 0.329 | 0.130 |
| Neura1LP | 0.084 | 0.181 | 0.035 | 0.117 | 0.273 | 0.048 | 0.101 | 0.190 | 0.064 | 0.148 | 0.271 | 0.101 |
| DRUM | 0.076 | 0.138 | 0.044 | 0.152 | 0.313 | 0.072 | 0.107 | 0.193 | 0.070 | 0.161 | 0.264 | 0.119 |
| NBFNet | 0.096 | 0.199 | 0.032 | 0.137 | 0.255 | 0.077 | 0.225 | 0.346 | 0.161 | 0.283 | 0.417 | 0.224 |
| RED-GNN | 0.212 | 0.385 | 0.114 | 0.203 | 0.353 | 0.129 | 0.179 | 0.280 | 0.115 | 0.214 | 0.266 | 0.166 |
| MaKEr* | 0.045 | 0.093 | 0.014 | 0.051 | 0.108 | 0.018 | 0.055 | 0.116 | 0.021 | 0.048 | 0.089 | 0.019 |
| INGRAM+ | 0.309 | 0.506 | 0.212 | 0.261 | 0.464 | 0.167 | 0.281 | 0.453 | 0.193 | 0.334 | 0.501 | 0.241 |
| HyRel | **0.394** | **0.574** | **0.299** | **0.305** | **0.502** | **0.205** | **0.321** | **0.520** | **0.222** | **0.348** | **0.541** | **0.263** |

| Model | FB-100 | | | FB-75 | | | FB-50 | | | FB-25 | | |
|---|---|---|---|---|---|---|---|---|---|---|---|---|
| | MRR | Hit@10 | Hit@1 | MRR | Hit@10 | Hit@1 | MRR | Hit@10 | Hit@1 | MRR | Hit@10 | Hit@1 |
| Neura1LP | 0.026 | 0.057 | 0.007 | 0.056 | 0.099 | 0.030 | 0.088 | 0.184 | 0.043 | 0.164 | 0.309 | 0.098 |
| DRUM | 0.034 | 0.077 | 0.011 | 0.065 | 0.121 | 0.034 | 0.101 | 0.191 | 0.061 | 0.175 | 0.320 | 0.109 |
| NBFNet | 0.072 | 0.154 | 0.026 | 0.089 | 0.166 | 0.048 | 0.130 | 0.259 | 0.071 | **0.224** | 0.410 | **0.137** |
| RED-GNN | 0.121 | 0.263 | 0.053 | 0.107 | 0.201 | 0.057 | 0.129 | 0.251 | 0.072 | 0.145 | 0.284 | 0.077 |
| INGRAM+ | 0.223 | 0.371 | 0.146 | 0.189 | 0.325 | 0.119 | 0.117 | 0.218 | 0.067 | 0.133 | 0.271 | 0.067 |
| HyRel | **0.282** | **0.463** | **0.188** | **0.277** | **0.433** | **0.196** | **0.178** | **0.333** | **0.101** | 0.210 | **0.420** | 0.114 |

| Model | WK-100 | | | WK-75 | | | WK-50 | | | WK-25 | | |
|---|---|---|---|---|---|---|---|---|---|---|---|---|
| | MRR | Hit@10 | Hit@1 | MRR | Hit@10 | Hit@1 | MRR | Hit@10 | Hit@1 | MRR | Hit@10 | Hit@1 |
| Neura1LP | 0.009 | 0.016 | 0.005 | 0.020 | 0.054 | 0.004 | 0.025 | 0.054 | 0.007 | 0.068 | 0.104 | 0.046 |
| DRUM | 0.010 | 0.019 | 0.004 | 0.020 | 0.043 | 0.007 | 0.017 | 0.046 | 0.002 | 0.064 | 0.116 | 0.035 |
| NBFNet | 0.014 | 0.026 | 0.005 | 0.072 | 0.172 | 0.028 | 0.062 | 0.105 | **0.036** | 0.154 | 0.301 | 0.092 |
| RED-GNN | 0.096 | 0.136 | 0.070 | 0.172 | 0.290 | 0.110 | 0.058 | 0.093 | 0.033 | 0.170 | 0.263 | 0.111 |
| INGRAM+ | **0.107** | **0.169** | **0.072** | 0.247 | 0.362 | 0.179 | **0.068** | 0.135 | 0.034 | 0.186 | 0.309 | 0.124 |
| HyRel | 0.091 | 0.165 | 0.059 | **0.255** | **0.389** | **0.187** | **0.068** | **0.138** | **0.036** | **0.191** | **0.316** | **0.125** |

## 5 EXPERIMENT

### 5.1 Datasets

The benchmarks used in our experiments are sourced from NELL-995[32], Wikidata68K[11], and FB15K237[26], configured for inductive settings. Each benchmark is subdivided into four datasets, where the percentages of triples with new relations are 100%, 75%, 50%, and 25%. All entities in the $\widetilde{\mathcal{G}_{\text{inf}}}$ of the 12 datasets are not observed during training. For instance, in NL-100, all triples in the inference graph involve unseen relations. The detailed information of the datasets and hyperparameters settings is presented in supplementary materials.

### 5.2 Evaluation Metrics and Baselines

To assess the performance of different methods, we employ three commonly used evaluation metrics: MRR, Hit@10, and Hit@1. MRR represents the mean reciprocal rank, while Hit@10 and Hit@1 indicate the proportion of correct answers ranked within the top 10 and top 1 among all candidates.

We compared the performance of our method with 16 methods on the task of inductive link prediction, including: INGRAM[14], MaKEr[4], GraIL[25], INDIGO[20], CoMPILE[21], BLP[7], QBLP[1], RAILD[11], CompGCN[28], NodePiece[9], NeuralLP[35], DRUM[22],

**Table 2: The performance of HyRel and baselines in inductive link prediction is evaluated on the NELL-995-v1 dataset, where all entities are unseen and the relations are seen.**

| Model | NELL-995-v1 | | |
|---|---|---|---|
| | MRR | Hit@10 | Hit@1 |
| GraIL | 0.499 | 0.595 | 0.405 |
| CoMPILE | 0.474 | 0.575 | 0.390 |
| SNRI | 0.419 | 0.520 | 0.330 |
| INDIGO | 0.521 | 0.595 | 0.495 |
| RMPI | 0.484 | 0.545 | 0.425 |
| CompGCN | 0.282 | 0.750 | 0.005 |
| NodePiece | 0.677 | 0.885 | 0.550 |
| Neura1LP | 0.547 | 0.785 | 0.400 |
| DRUM | 0.536 | 0.760 | 0.400 |
| BLP | 0.169 | 0.470 | 0.055 |
| QBLP | 0.326 | 0.545 | 0.230 |
| NBFNet | 0.613 | 0.875 | 0.500 |
| RED-GNN | 0.544 | 0.705 | 0.470 |
| RAILD | 0.052 | 0.205 | 0.000 |
| MaKEr* | 0.552 | 0.725 | 0.480 |
| INGRAM+ | 0.739 | 0.895 | **0.660** |
| HyRel | **0.750** | **0.910** | **0.660** |

NBFNet[38], RMPI[10], SNRI[33] and RED-GNN[37]. Among them, results on FB and NL datasets cannot be given due to scalability issues with GraIL, CoMPILE, SNRI, INDIGO, RMPI, and MaKEr. InGram, MaKEr and RMPI achieve inductive link prediction where entities and relations are unseen. For fair comparison, we set the candidate entities for all methods to be all entities in $\mathcal{T}_{inf}$. The experimental results and experimental settings of some methods are derived from INGRAM[14].

## 5.3 Main Results of Inductive Link Prediction

From Table 1, it can be observed that our method outperforms the baseline models in almost all of the 12 datasets.

**Fully Inductive Inference.** Specifically, concerning the most challenging task: the fully inductive inference task on datasets NL-100, FB-100, and WK-100, where both relations and entities are entirely unseen, our method significantly outperforms all baseline methods. In NL-100 and FB-100, our approach achieves noticeable improvements of 8.5% and 5.9% in terms of MRR metric, respectively, compared to the previous best results. Moreover, our model also demonstrates competitive performance on WK-100.

**Semi Inductive Inference.** Apart from NL-100, FB-100, and WK-100, the other 9 datasets consist of both seen and unseen relations. It is evident from these 9 datasets that our method outperforms other baseline models significantly on 8 datasets, except for FB-25 where NBFNet[38] performs comparably to ours. Actually, simple rules exist among the seen relations in FB-25, making methods rely on rule-based predictions, more suitable for this dataset. Rule-based prediction methods require capturing rules among seen relations, whereas our method does not focus on fixed patterns; instead, we emphasize general cases. Our model aims to ensure generalization

performance by adopting a more universally applicable approach to embedding acquisition.

**Inductive Inference With Seen Relations.** We conducted experiments under the setting where all relations are seen. We used the existing benchmark dataset NELL-995-v1[14] for comparison. The experimental results are shown in Table 2, indicating that our model also exhibits the strongest inductive link prediction capability when all relations are seen.

These strong performances from all comparative experiments indicate the effectiveness of our HyRel in addressing inductive link prediction tasks. HyRel adapts to various inductive settings, demonstrating superior accuracy in embedding for both fully inductive and semi-inductive KG.

**Table 3: Ablation Studies of HyRel.**

| Model Variants | NL-100 | | NL-75 | |
|---|---|---|---|---|
| | MRR | Hit@1 | MRR | Hit@1 |
| w/o Relation positions | 0.324 | 0.209 | 0.267 | 0.155 |
| w/o Affinity | 0.374 | 0.288 | 0.286 | 0.183 |
| w/o HyRelGat | 0.251 | 0.136 | 0.204 | 0.103 |
| w/o Dual CL | 0.356 | 0.265 | 0.290 | 0.195 |
| w Common CL | 0.302 | 0.207 | 0.250 | 0.141 |
| HyRel | **0.394** | **0.299** | **0.305** | **0.205** |

| Model Variants | FB-100 | | FB-75 | |
|---|---|---|---|---|
| | MRR | Hit@1 | MRR | Hit@1 |
| w/o Relation positions | 0.227 | 0.143 | 0.217 | 0.128 |
| w/o Affinity | 0.253 | 0.169 | 0.258 | 0.168 |
| w/o HyRelGat | 0.198 | 0.128 | 0.168 | 0.099 |
| w/o Dual CL | 0.249 | 0.165 | 0.252 | 0.166 |
| w Common CL | 0.266 | 0.168 | 0.244 | 0.159 |
| HyRel | **0.282** | **0.188** | **0.277** | **0.196** |

## 5.4 Ablation Study

We conducted ablation experiments to demonstrate the significance of each module. The results of the ablation experiments are presented in Table 3, where each ablation variants leads to a performance decrease. Specifically, we trained our model under the following ablation settings: (1) w/o relation positions: disregarding relation positions by integrating the views under four positions into a single graph, with embedding updates performed on the integrated graph. (2) w/o affinity: removing omitting affinity weights during embedding updates for each view. (3) w/o HyRelGat: ablating the four views of relations attention module. (4) w/o dual CL: ablating the dual views contrastive constraints module. (5) w common CL: replacing the dual views contrastive constraints approach with common contrastive learning, where only nodes of the same relation across different views are pulled together, while all other targets are pushed apart. The ablation results indicate the importance of different components, and their joint modeling yields the best results. The removal of the HyRelGat module resulted in a noteworthy performance drop. This is because the variant cannot utilize the relative positions and affinities of relations for learning

at the same time. Thus the substantial performance degradation validates the rationale behind our approach.

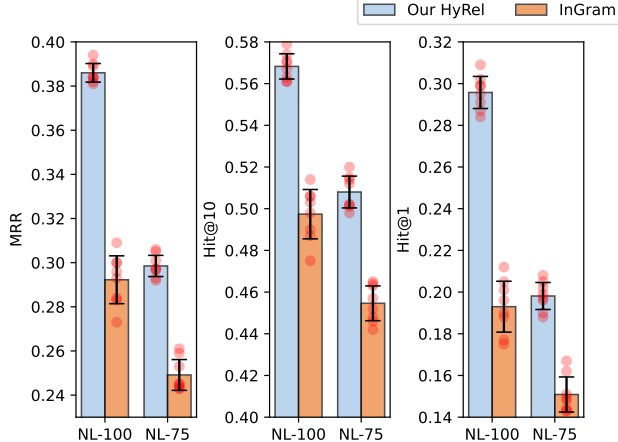

**Figure 4: Comparison of the magnitude of randomness between HyRel and INGRAM.**

## 5.5 Model Randomness

Some existing inductive link prediction methods alter the training regimes to enhance model generalization, leading to randomness in model outcomes. This situation also applies to how the training set is divided in each epoch. We compare HyRel with INGRAM, both of which are trained using this method of dividing the training set. As shown in Fig. 4, our method exhibits significantly smaller error ranges across all three metrics on the NL-100 and NL-75 datasets compared to INGRAM. This indicates that our model's good performance is not a result of random chance but rather its ability to generate embeddings for unseen entities and relations more reasonably.

**Table 4: Performance comparison of different views as central views on the NL-100 dataset.**

| Model | NL-100 | |
|---|---|---|
| | MRR | Hit@1 |
| HyRel-($t$-$t$) | 0.384 | 0.301 |
| HyRel-($h$-$t$) | 0.396 | 0.306 |
| HyRel-($t$-$h$) | 0.387 | 0.303 |
| HyRel-($h$-$h$) | 0.394 | 0.299 |

## 5.6 Experimental results of different central views.

In practice, the choice of central view in four views has minimal impact on performance, as shown in Table 4 with relevant experimental results. It can be observed that the impact of different central

views on the experimental results is relatively small. Although there are fluctuations, they are within an acceptable range due to the presence of randomness.

## 5.7 Case Studies

Relation pairs outlined with dashed ellipses in Fig. 5 represent adjacent nodes with high affinities. It can be observed that relations with significant semantic correlations, such as *Athlete_plays_sport* and *Athlete_plays_in_league* exhibit relatively smaller distances in the visualization space compared to other nodes. For relation $r_1$, $r_0$ and $r_{52}$ represent the relations with the highest and lowest affinities, respectively. For relation $r_8$, $r_9$ and $r_{37}$ are relations with comparable affinities. Fig. 5 clearly demonstrates that the distances of relation embeddings are consistent with the magnitudes of their affinities. The distance between $(r_9, r_8)$ and $(r_{37}, r_8)$ is very close, while the distance between $(r_1, r_0)$ is much smaller than that of the zero-affinity relation pair $(r_1, r_{52})$. The embedding visualization proves semantic representations learned by our HyRel model are reasonable, which affirms the effectiveness of HyRel embedding unseen relations.

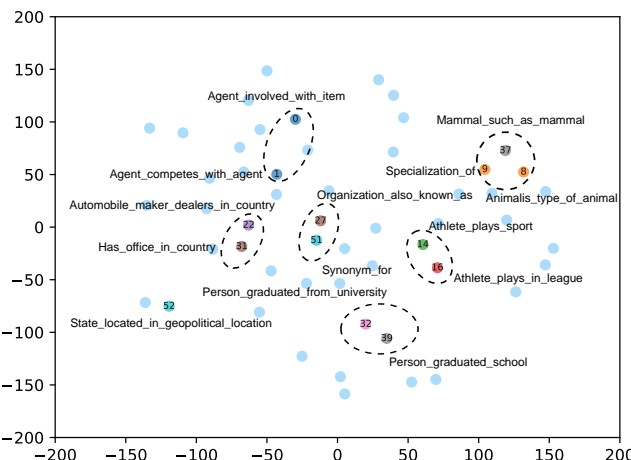

**Figure 5: Visualization of HyRel's relation embeddings using T-SNE on the NL-100 dataset.**

## 6 CONCLUSION

We introduce HyRel, a pioneering approach tailored for addressing inductive link prediction adopting the hyper-relation structure. HyRel operates independently, void of dependencies on pre-trained language models or external sources. Emphasizing the acquisition of robust reasoning capabilities, HyRel leverages the hyper-relation structure to effectively capture graph structural intricacies. This empowers HyRel to generate precise embeddings for unseen relations and entities. Moreover, the proposed dual views contrastive constraints are imposed across views, which alleviates semantic confusion of relations. Extensive experiments across diverse inductive settings validate HyRel's superior performance. HyRel offers a promising avenue for addressing the challenges of knowledge extrapolation in evolving KG environments.

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
