# OpenReview forum: "Generalize to Fully Unseen Graphs: Learn Transferable Hyper-Relation Structures for Inductive Link Prediction"
_acmmm.org/ACMMM/2024/Conference — MM2024 Poster_

### Official Review · Reviewer_DheK · 2024-05-08

**Rating:** 5
**Confidence:** 3

**Summary:**

This paper proposes a novel HyRel model to enhance the generalization ability of inductive link prediction in unseen graphs. HyRel improves the model's performance on completely unseen graphs by learning a globally transferable graph structure, particularly learning a hyper-relational structure consisting of relational positions and affinities. The authors mainly introduce a novel perspective that focuses on learning the hyper-relational structure inherent in graphs, enabling the learned knowledge to be transferred to any unseen graph. For the first time, affinities between relations at different relative positions are incorporated into inductive link prediction. Aggregate finer semantic information by exploiting the semantic differences between different relative positions. Dual-view contrast constraints are proposed to alleviate the semantic confusion of relations and enhance the robustness of transferable structural knowledge.

**Strengths:**

This article presents the practical application of the HyRel model, extracting the hyper-relational structure from the global structure of a given graph. By considering the affinities between relations at different relative positions, four views are constructed, each representing a different relative position of the relationship. Affinity is defined by calculating the frequency of shared entities between relationships at different relative positions. The authors proposed the HyRel-Gat mechanism for generating high-quality relation embeddings. Dual-view contrast constraints are designed to alleviate the semantic confusion of relationships and enhance the robustness of transferable structural knowledge, thereby inspiring new possibilities in the field of inductive link prediction.

**Limitations:**

Although the paper is well organized and logical, some parts must be changed and complemented.

 First, it is hoped that the author can provide a more detailed description of the experimental settings, including the data set used, data preprocessing steps, hyperparameter selection and other information, to increase the repeatability and credibility of the experimental results.

Second, to improve the model's interpretability, it is recommended that a more in-depth explanation of the inner workings of the HyRel model be provided, including the role of key components and how they affect the final prediction results.

**Suitability:**

2

---

### Official Review · Reviewer_qVrL · 2024-05-23

**Rating:** 4
**Confidence:** 3

**Summary:**

This paper proposes a called HyRel for inductive link prediction. HyRel aims to enhance generalization ability by learning the global transferable structure of graphs, specifically focusing on the inherent hyper-relation structure. The hyper-relation structure is independent of specific entities, relations, or features, allowing for knowledge transfer to any unseen graphs. Experiments on multiple benchmarks validate the efficacy of HyRel.

**Strengths:**

1. This paper is easy to follow. The motivation and architecture of HyRel are clearly illustrated.

2. The results across multiple benchmarks are promising, significantly outperforming the baselines.

3. Extensive experiments are conducted to further analyze the proposed method.

**Limitations:**

1. As the paper mostly follows INGRAM’s experiment settings, it would be valuable to include the performance of HyRel on other benchmarks such as FB15k-237 and WN18RR proposed by GraIL.

2. Honestly, I think this paper lacks relevance to multi-media or mutli-modality, please, may I ask, why the authors submit it to MM?

typo: Line 132: 'are' is missing.

**Suitability:**

1

---

### Official Review · Reviewer_v5Qe · 2024-05-26

**Rating:** 4
**Confidence:** 3

**Summary:**

The paper presents a method for inductive link prediction in knowledge graphs by learning transferable hyper-relation structures. It introduces the concept of hyper-relation structure to represent complex relationships between entities and proposes a model called HyRelGat that updates relation representations using graph attention networks. The model employs dual views contrastive constraints to enhance generalization across unseen graphs. Extensive experiments on various datasets demonstrate the effectiveness of the proposed approach in predicting links in fully unseen graphs, outperforming several baseline models. The study also includes ablation analysis, randomness analysis, and case studies to validate the model's performance and robustness.

**Strengths:**

The paper presents a novel approach for inductive link prediction by learning transferable hyper-relation structures, which is particularly useful for generalizing to unseen graphs. It introduces the HyRelGat model that effectively updates relation representations and applies dual views contrastive constraints for improved performance. The ablation study demonstrates the significance of each component, and case studies provide insights into the model's behavior. Overall, the method shows promising results in enhancing graph representation learning for relational data. Particularly, the experiments are sufficient enough to support the reliability of the proposed method.

**Limitations:**

Although it is appreciated that this paper can provide many benchmarking results between the proposed method and the available state-of-arts methods, I still suggest the author to consider categorize the available methods and select the typical several ones for benchmarking test.

Regarding "From Table 1, it can be observed that our method outperforms the baseline models in almost all of the 12 datasets", I have similar concern. I still suggest to give the readers more insights about the datasets. Why the author selects so many datasets for comparison? What is the point of each dataset concerning? Whether or not the proposed method can be proved to work well on this point?

In session 6. Conclusion, "HyRel operates independently, void of dependencies on pre-trained language models or external sources." I suggest the author to provide the proof that using the proposed HyRel method, it can work well even combining the pre-trained language models. In my understanding, it is an unvoidable situation nowadays.

**Suitability:**

2

---

### Official Review · Reviewer_kb9F · 2024-05-27

**Rating:** 4
**Confidence:** 3

**Summary:**

This paper introduces a hyper-relation aware multi-views model  HyRel for semi- and fully-inductive link prediction task. HyRel first constructs hyper-relation graphs with 4 views, and then update relation representation and entity representation based on these 4 views. It also propose a novel contrastive learning imposing contrastive constraints on 4 views. Experiments on existing benchmark of semi- and fully-inductive link prediction benchmarks show that HyRel performs well, especially on fully-inductive setting.

**Strengths:**

1. This paper focus on an interesting and challenging task, inductive link prediction, especially in fully-inductive setting.
2. The method proposed in this paper has a  quite strong generalizability, that could be used for diverse inductive setting with different new-relation ratio.
3. The experiment results support and prove the effectiveness of the proposed method.

**Limitations:**

1. The code is not available with the given link.
2. In the experiments of inductive inference with seen relations, why only NELL-995-v1 is chose for this experiments? as there are a lot of version of data are available.

**Suitability:**

2

---

### Meta-Review · Area_Chair_XEQG · 2024-07-03

**Recommendation:** Accept (Poster)
**Confidence:** 4

**Metareview:**

The paper presents an innovative approach to inductive link prediction, particularly in a fully inductive setting, through the HyRelGat model. The strengths highlighted by the reviewers include the method's generalizability across diverse inductive settings, effective update of relation representations, and dual views contrastive constraints that enhance performance. The ablation studies and case studies further substantiate the model's effectiveness, with results significantly outperforming baselines across multiple benchmarks. While the paper is well-organized and provides clear illustrations of motivation and architecture, it faces some limitations such as the need for broader benchmark comparisons and more detailed experimental settings to improve reproducibility and interpretability. Despite minor concerns about the code availability and dataset choices, the overall contributions and promising results support the acceptance of this paper.